# Understanding Antimicrobial Use Contexts in the Poultry Sector: Challenges for Small-Scale Layer Farms in Kenya

**DOI:** 10.3390/antibiotics10020106

**Published:** 2021-01-22

**Authors:** Stella Kiambi, Rosemary Mwanza, Anima Sirma, Christine Czerniak, Tabitha Kimani, Emmanuel Kabali, Alejandro Dorado-Garcia, Suzanne Eckford, Cortney Price, Stephen Gikonyo, Denis K. Byarugaba, Mark A. Caudell

**Affiliations:** 1Food and Agriculture Organization of the United Nations, Nairobi PO Box 30470, Kenya; tabitha.kimani@fao.org (T.K.); stephen.gikonyo@fao.org (S.G.); mark.caudell@fao.org (M.A.C.); 2State Department of Livestock Production, Livestock Fisheries and Cooperative, Ministry of Agriculture, Nairobi PO Box 34188, Kenya; rnmwanza@gmail.com; 3State Department of Veterinary Services, Livestock Fisheries and Cooperative, Ministry of Agriculture, Nairobi PO Box 34188, Kenya; asirma@kilimo.go.ke; 4Food and Agriculture Organization of the United Nations, 00153 Rome, Italy; christine.czerniak@fao.org (C.C.); emmanuel.kabali@fao.org (E.K.); alejandro.doradogarcia@fao.org (A.D.-G.); cortney.price@fao.org (C.P.); 5Department for Environment, Food and Rural Affairs, Veterinary Medicines Directorate, Surrey KT15 3LS, UK; s.eckford@vmd.gov.uk; 6College of Veterinary Sciences, Makerere University, Kampala PO Box 7022, Uganda; byarugabadk@yahoo.com

**Keywords:** antimicrobial use, poultry, Kenya, biosecurity, antimicrobial resistance, mixed-methods, antibiotic, drug, behavior

## Abstract

The poultry sector contributes significantly to Kenya’s food and economic security. This contribution is expected to rise dramatically with a growing population, urbanization, and preferences for animal-source foods. Antimicrobial resistance is putting the poultry sector in Kenya—and worldwide—at risk of production losses due to the failure of medicines for animal (and human) health. The emergence and spread of antimicrobial resistance has been linked to overuse and misuse of antimicrobials in poultry and other sectors. Previous studies have documented poultry farmer antimicrobial use but without systematic consideration of the contexts (i.e., drivers) as important targets for behavior change, particularly in low- and middle-income countries. To improve understanding of antimicrobial use patterns in poultry systems, we conducted a mixed-methods knowledge, attitudes, and practices study of 76 layer farms in Kiambu County; Kenya. We found that commonly used antibiotics were often labeled for prophylactic, growth promotion, and egg production improvement purposes. Antimicrobial use was also motivated by the presence of diseases/disease symptoms, most of which could instead be managed through infection prevention measures. The results suggest that improving vaccination and biosecurity practices on farms and engaging with drug-makers to ensure proper labeling and marketing of antimicrobial drugs may represent important areas of opportunity for social behavior change communication and/or behavioral science interventions (i.e., nudges) to reduce disease burdens and promote prudent antimicrobial use. We conclude our findings with suggestions for further research into the behavioral insights at play in these scenarios to fuel future intervention development.

## 1. Introduction

The poultry sector is a cornerstone of food and economic security in Kenya. In 2014, the contributions of poultry and eggs to Kenya’s agricultural gross domestic product were estimated at 1.3% (USD 46.16 million) and 2.9% (USD 103.05 million), respectively [1]. Poultry and egg production are expected to continue rising dramatically in Kenya (and in other low- and middle-income countries (LMICs)) in response to population growth, urbanization. and an increased preference for animal-source foods [2]. The Food and Agriculture Organization of the United Nations (FAO) has predicted that demand for poultry in Kenya will rapidly increase within the next 30 years, and that demand for eggs will increase from approximately 89,000 tons per year in 2010 to 238,000 tons per year by 2030, and to almost 537,000 tons per year by 2050 [3].

The poultry industry in Kenya—and worldwide—is under threat from the emergence and spread of drug-resistant infections due to antimicrobial resistance. These infections are expected to result in significant production losses and the widespread failure of medicines for animal and human health unless action is taken to better control antimicrobial resistance (AMR) [4,5]. AMR—the ability of microorganisms such as bacteria, fungi and viruses to tolerate the antimicrobial medicines used to treat diseases caused by these pathogens—leads to therapeutic failures with downstream impacts on animal welfare, food safety, food security, and public health. AMR can be transmitted between people, animals, and the environment through direct contact and indirectly through the environment and food supply. Studies in Kenya and other LMICs indicate that AMR is already spreading through poultry systems [5,6,7]. A study conducted in Kenya to estimate AMR patterns in various bacteria isolated from chickens found a high prevalence of resistance to commonly used antibiotics, including ampicillin (76%), tetracycline (71%), sulphamethoxazole (70%), and co-trimoxazole (66%) [8]. In another Kenyan study, approximately 40% of *Salmonella typhimurium* and *Salmonella enteritidis* isolates from seemingly healthy animals and animal products showed resistance to commonly used antibiotics, making these drug-resistant bacteria a public health threat [9].

The emergence and spread of AMR in poultry, as in other production systems and public health contexts, has been linked to the overuse and misuse of antimicrobial drugs [10]. Data suggests that antimicrobial use (AMU) in the poultry sector is likely to be higher than in other production systems. A review of studies conducted in LMICs found that antimicrobial use was notably higher in chickens at 138 doses/1000 animal-days versus 40.2 doses/1000 animal-days for pigs and 10 doses/1000 animal-days for cattle [11]. In addition, antimicrobial consumption for poultry is expected to increase significantly over the next 30 years, particularly in LMICs, where intensification and resulting animal densities are expected to dramatically increase the volume of therapeutic antimicrobial use in response to endemic diseases. Volume increases are also expected for the nontherapeutic use of antimicrobials, including for growth promotion, prophylaxis and metaphylaxis [12,13,14,15].

The high consumption of antimicrobial use in poultry systems, combined with projected increases, is additionally concerning given evidence of non-prudent use of antimicrobials in these systems, as indicated by studies within LMICs [7,16,17,18,19,20,21,22,23,24,25]. Antimicrobials are usually purchased without prescriptions, risking use of the wrong drug or the purchase of a drug that needs to be held in reserve as a last resort [26]. These medicines are also typically self-administered (i.e., farmers administer antibiotics to animals themselves without veterinary intervention). This has been documented for poultry farmers in Ghana [17,20], Kenya [22], Tanzania [23], and Zambia and Zimbabwe [19]. Recommended antimicrobial treatment regimens are often reportedly disregarded as well [17]. Poultry farmers have also been shown to disregard antimicrobial withdrawal periods—the period of time before slaughter when treatments for the animal must cease in order to effectively eliminate the drug from the animal’s system before it is consumed [7,19,20,25].

To develop a better understanding of antimicrobial use patterns in poultry systems in Kenya, we conducted a mixed-methods study of layer farms in Kiambu County, Kenya. The objective of this study was to document patterns of antimicrobial use as potential targets for future interventions and relate these patterns to major domains of poultry production, including the sourcing of day-old chickens, disease burdens and biosecurity practices. For this investigation, we used a “bottom-up approach” to understanding antimicrobial use [19] by combining quantitative data from knowledge, attitudes, and practices (KAP) surveys with qualitative interviews to provide additional context. We found that antimicrobials commonly used by layer farmers are labeled for prophylactic and growth promotion purposes and that antimicrobial use is driven by practices that could be addressed through infection prevention measures such as hygiene, sanitation, and biosecurity practices as well as vaccinations. The results of our investigations were examined in order to suggest behavioral change interventions that may promote prudent antimicrobial use among Kenyan layer farmers.

## 2. Materials and Methods

### 2.1. Study Area and Population

The study was conducted among layer farmers in Gatundu North sub-county, Kiambu County of Kenya (see Figure 1), between December 2018 and March 2019. Kiambu County bounds the northern border of Nairobi, the capital city of Kenya. This county was chosen as the target for this study because it has the highest population of commercial chicken compared to other counties, totaling 2,538,359 according to 2016 livestock data, with 42% of these chickens purposed as layers, 33% as broilers, and the remaining ~25% classed as indigenous (i.e., varieties adapted to local environmental conditions, also referred to as “traditional”, “scavenging”, “backyard”, and “village” chickens). Within Kiambu Country, Gatundu North sub-county has the highest reported number of layers with approximately 241,500 [27]. Kiambu was chosen because prior research showed that antibiotics are widely used by farmers and acquired without prescription from agrovet shops [22]. Finally, poultry production and other types of agriculture are important to local livelihoods and the economy of Kiambu; therefore, this area is particularly vulnerable to the negative impacts of antimicrobial resistance.

### 2.2. Study Design

An interdisciplinary research team comprised of animal health experts and social scientists from the FAO, the Ministry of Agriculture, Livestock, Fisheries and Irrigation, and sub-county animal healthcare workers collaborated on study design and implementation. During the first round of data collection (December 2018–January 2019), the research team collected qualitative data using focus group discussions (FGDs) and in-depth interviews (IDIs). Six mixed-gender FGDs were conducted with farmers, six IDIs with agrovets, and six IDIs with sub-county veterinary officers and assistant veterinary officers. FGDs and IDIs were concentrated around twelve major themes relating to AMU and AMR including farm management and economic practices, disease histories, and marketing (see Appendix A for a list of all themes and prompts to probe themes). Participants in FGDs and IDIs were selected with input from sub-county animal healthcare workers to balance gender, farm scale, and length of time in poultry production.

Thematic analysis of qualitative interviews was used to develop a KAP survey instrument, which was deployed during the second round of data collection (February 2019–March 2019). The KAP survey included a broad range of questions on demographics, livelihood, health, hygiene and biosecurity topics relating to factors that could promote AMU, suboptimal AMU practices, and AMR (see Appendix A for KAP questionnaire). The sampling frame for the KAP survey was generated by research assistants conducting a door-to-door census of layer farmers living in Gatundu North. Interviews lasted an average of one hour. All stakeholders provided informed consent to participate through signature or thumbprint. The survey was administered in English, Kiswahili, or Kikuyu, with the respondent selecting his or her preferred language. KAP surveys were administered by six local research assistants using tablets with the Kobo Collect^®^ Cambridge, MA, USA application. All assistants participated in a four-day training workshop followed by a KAP piloting exercise to evaluate and standardize the approach.

Several methods were used to collect antimicrobial use data. First, use data was collected by creating posters that included pictures of all locally available drugs. Pictures on the posters were taken at agrovet drugs shops in the communities where the respondents resided. Enumerators would give the respondents these posters and have them point out the drugs they used. See Appendix A for an example of these posters. Second, enumerators asked respondents to bring any drugs kept at the house/farm. If these drugs were not already chosen from the poster-exercise, then they were recorded. Finally, enumerators asked respondents if they could recall any other drugs used that were not included on the poster or that they were not keeping at home currently.

### 2.3. Ethical Consideration

This study was approved by the AMREF Health Africa Ethics and Scientific Review Committee (AMREF-ESRC P551/2018). The research was also approved by the Institutional Animal Care and Use Committee of KALRO-Veterinary Science Research Institute, Muguga, upon compliance with all provision vetted under and coded: KALRO-VSRI/IACUC016/28092018.

### 2.4. Data Analysis

To explore the factors associated with AMU, we used Chi-square tests to correlate whether a respondent reported giving antimicrobials to birds in a “typical month of production” (Yes = 1, No = 0) with self-reported practices regarding vaccinations and biosecurity measures, treatment of sick birds, common types of diseases, and attitudes towards prudent antimicrobial use. We also present details on the types of antimicrobials used as recalled from participants through using drug posters, household inventories, and self-recall (see 2.2. Study Design above). To obtain details on the active ingredients and manufacturer’s indications for use of the recalled medicines, we used the medicine labels and, if this information was not available, the trade names were searched over the internet. Data cleaning, coding, and analysis were done in Stata 16 [28].

### 2.5. Presentation of Results

In the Results and Discussion section, we synthesize qualitative (IDI, FGDs) and quantitative (KAP) data to examine patterns of AMU in Kiambu layer farms. We first provide demographic statistics on our KAP survey respondents followed by discussion of the major domains of production, including management of poultry diseases, vaccination, biosecurity, and patterns of AMU. In the Discussion, we use qualitative data from our follow-up assessments (April 2019) to explain initial qualitative and quantitative findings.

## 3. Results and Discussion

### 3.1. Descriptive Statistics of KAP Survey Respondents

A total of 76 households were interviewed. Efforts were made to ensure equal representation of women and so the survey respondents were split along gender lines with 52% of respondents being female. Farm owner gender, however, was highly male-skewed, with over 97% of farm owners being male. Respondents were on average 50 years old, had completed secondary school education, and had on average ten years of experience in keeping layers. Farms had an average of three poultry houses (min = 1, max = 11) holding an average of 1000 layers total (min = 90, max = 19,000). Other animals kept on the farms included broilers (avg. ≈ 6, min = 0, max = 100), dairy cattle (avg. ≈ 3, min = 0, max = 16), dogs (avg. ≈ 1, min = 0, max = 5), sheep (avg ≈ 1, min = 0, max = 5), and goats (avg. ≈ 1, min = 0, max = 8). See Appendix A for demographics and farm level information, respectively.

#### 3.1.1. Management and Control of Poultry Diseases

FGDs with farmers revealed extensive knowledge of a wide range of symptoms and diseases that affected their layer flocks. Over 20 common symptoms were listed in our FGDs and included in our KAP survey questions. Results from the KAP surveyed revealed that abnormal eggs and a drop in egg production were reported by 92% and 87% respondents, respectively. Snoring was reported by over 70% of households. Diarrhea (bloody, yellow, and white) was reported by over 50% of respondents. See Appendix A for a full list of recalled symptoms. To collect disease data, enumerators asked respondents what diseases their birds had been “diagnosed” with, although we could not verify whether diagnosis derived from samples sent to a laboratory, a qualified health professional, attendants at agrovet shops, or self-diagnosis. With this caveat highlighted, there were several trends across respondents. The top diseases mentioned included: Newcastle disease, locally known as “*kihuruto*”, chronic respiratory disease (CRD) (“*mung’oroto*”), coccidiosis, egg peritonitis, and infectious bursal disease (“*Gumboro*”). Respondents mentioned that the most common killer of their chicks up to six weeks of age was *Gumboro*, which they believed was transmitted at the hatchery and was therefore one of the reasons that they would keep changing the source of their day old chicks. Our survey results were coherent with the qualitative data indicating coccidiosis (64%) and CRD (63%) were two of the most commonly reported diseases (Figure 2). These results are consistent with surveys of commercial poultry farmers in Tanzania [7]. Only 3% of households reported not experiencing any disease whatsoever. Across a normal production cycle, the median reported mortality was 50 birds with a first quartile of 25 birds and a third quartile of 100 birds.

#### 3.1.2. Vaccination Practices

Farmers explained that even though they followed the vaccination schedule given by the hatchery as best as they could, they still experienced outbreaks, particularly for Newcastle Disease. One of the farmers said, “*Kihuruto will infect birds even if you have vaccinated, so we just vaccinate for the sake of following the given advice, but sometimes we forget to vaccinate*”. However, one of the animal health professionals at an agrovet said that most farmers usually vaccinated their flocks too late for the vaccine to be effective. He said, “*Some farmers vaccinate late, or will not vaccinate until they hear of outbreaks in the neighborhood*”. Despite these occurrences, self-reported vaccination rates were high, with all farmers reporting having vaccinated against Gumboro (100%), Newcastle disease (100%), Newcastle combined with Gumboro (95%), and Fowl Pox and Fowl Typhoid (both 95%), with few vaccinating for Marek’s Disease (4%).

#### 3.1.3. Biosecurity Practices for Infection Prevention

Respondents recognized the need for biosecurity measures to minimize the spread of infections. For example, farmers mentioned that they observed human traffic restrictions where they only allowed a few people in the poultry houses and this was limited mostly to those concerned with feeding and egg collection. However, results from our KAP survey suggested some reluctance in investing in biosecurity measures. Only about 51% participants had footbaths at the entrance of their layer houses. Of those who indicated having a footbath, 51% described their footbaths as sponges/mats soaked in plain water (without soap or disinfectant), 33% had mats soaked in disinfectants, while 13.4% had concrete troughs with plain water, and 2.6% had concrete troughs with disinfectant. Most households (≈49%) reported cleaning and replenishing their footbaths once or twice per week, others did so every day (≈28%), and the smallest percentage did so monthly (≈3%). About 21% of the farmers reported never changing footbaths, only filling them up after they dried out. Those who did not have footbaths said they sprayed the chicken house twice a month with the disinfectant Coco-benzyl (dimethyl-ammonium chloride). The spraying was said to be done while the birds were still in the houses and was applied on the floor, walls, in the air, and on the walls outside the chicken house. While not a recommended practice, the farmers also said that they gave some of the disinfectant to chickens in their drinking water as a way of preventing diseases.

For personal protective equipment (PPE), most households reported having various PPE including boots (≈75%), overalls (50%), gloves (51%), and masks (58%). However, a majority of the households did not have specific PPE per layer house. Around 10% of the participants cleaned their PPE after every use in the layer house, while 30% cleaned after a few uses and 27% cleaned them only when they became visibly dirty.

The cleaning of drinkers was done twice a week by most of the farmers (≈69%), while the rest reported cleaning once or twice a month (≈19%) or at the end of the production cycle (≈11%). Only 1% of the farmers reported that they never cleaned the drinkers. Cleaning was done by most respondents using plain water (43%), followed by the use of water and soap (≈30%) and water with disinfectant (≈18%), while the rest (≈9%) used a water, soap, and disinfectant combination. For feeders, most of the farmers (≈72%) reported cleaning the feeders only after the end of the production cycle. A few reported that they cleaned once a month (≈13%) and others said they cleaned once or twice a week (≈3%). About 12% of the farmers said they did not clean the feeders at all. Of those who cleaned, a majority (≈42%) used water, disinfectant and soap. About 28% of the farmers reported using water and disinfectant, 26% used water and soap, and about 4% of the farmers reported cleaning with plain water.

Almost all households had layer houses that met standardventilation criteria with most having walls of at least 50 cm height (≈92%), two walls with wire mesh openings (≈97%), and adequate space between the roof and wall (≈85%). However, due to space constraints all layer houses were close to other houses occupied by humans or other animals (<5 m) and none had any temperature measuring facilities in the layer houses.

#### 3.1.4. Qualitative Data from Animal Health Professionals

Qualitative and quantitative data from layer farmers on their vaccination and biosecurity practices were consistent with interviews among animal health professionals. Animal health professionals overwhelmingly said that poultry diseases, and by extension antimicrobial use, were caused by poor biosecurity practices among the farmers. For example, one of the animal health professionals said that mixing of indigenous chickens and commercial hybrid flocks was a common occurrence in the farms that he visited and that these farms rarely vaccinated the indigenous chickens. He commented that “*Indigenous chickens are almost never vaccinated and scavenge outside the chicken pen so can transmit disease to the commercial birds*”. Other poor biosecurity practices mentioned included a lack of footbaths, sharing of poultry equipment between farmers, failure to observe floor space requirements, and not considering the wind direction during poultry shed construction, leading to overcrowding and poor ventilation for the latter two practices, respectively. Other issues noted by animal health professionals were allowing visitors into the poultry shed, mismanagement of feed formulations that led to compromised immunities, and rodents/snakes entering the poultry house.

#### 3.1.5. Commonly Used Medicines

Across the 76 households, farmers reported commonly using about 45 different types of medicines with over 62% (28) identified as antimicrobials. The others were plain multivitamins, probiotics, and dewormers. In Table 1, we list the antibiotics reported by 10% or more of households. Most of these products were a combination of broad-spectrum antibiotics and multivitamins. We define a “broad spectrum drug” as a medicine that contains one or more antibiotics that act against a diverse range of pathogenic bacteria. Respondents of FGDs mentioned several medicines as the most preferred, which they referred to as the “*magic drugs*”. These included Fosbac^®^ (fosfomycin and tylosin), Tylodox^®^ (Tylosin tartrate 100 mg and Doxycycline hyclate 200 mg), Limoxil^®^ (Oxytetracycline), Tylodoxine^®^ (Doxycycline and Tylosin Tartrate), and Tylosine 75^®^ (Tylosin tartrate eq. 750 000 I.U. and 750 mg Tylosin). One of the female farmers said that “*all these are wonder drugs for treatment of mung’oroto* [CRD]”. High reliance on tetracylines is consistent with findings from poultry farmers in Tanzania [7] and Ghana [20,25], although these studies did not report use of tylosin.

#### 3.1.6. AMU and AMU-Related Behaviors

In the KAP survey, a large majority of farmers reported observing prudent AMU practices, including never or rarely using a larger or smaller dose than recommended, stopping treatment early, or using expired medicines or medicines used for humans (see Table 2). These results, specifically giving the full treatment, contrast with a study in poultry farmers in Ghana where only 63% of farmers completed the full treatment course [17]. However, there was some evidence of non-prudent use with about 90% of respondents treating all birds when one bird became sick and about 75% saying they used antimicrobials to prevent diseases.

For AMU withdrawal practices, only about 47.5% of the farmers reported having heard about a “withdrawal period”. These results are consistent with studies conducted among poultry farmers in Tanzania [7], Ghana [19,20,25], Zambia [19], and Zimbabwe [19]. Assessment of responses from FGD indicated that a disregard for withdrawal period was due to some respondents not knowing the food safety or public health implications of not observing the withdrawal period. Respondents had different strategies to deal with eggs from layers undergoing treatment and during the withdrawal period. Most of the eggs were reported to be sold (≈89.5%) followed by consumption at home (≈40.8%). Other ways of dealing with this included sharing with neighbors or friends (≈18.4%) and feeding them to household dogs (≈14.5%). Some were also said to be thrown into the pit latrines (≈2.6%) and into the compost pits (≈1.3%).

### 3.2. Analyses of the Factors Associated with AMU

Chi-squared tests indicated that practices regarding vaccinations, biosecurity, burdens of diseases, and observance of prudent-use practices were largely not significantly correlated with antimicrobial use. See the “Variables entered into Chi square analysis” dataset in the Appendix A. for a description of all variables entered in the Chi-squared analysis. Respondents were significantly more likely (at the *p* < 0.05 level) to report using antimicrobials if they owned gloves *X*^2^ (1, *N* = 76) = 4.6587, = 0.031) or overalls *X*^2^ (1, *N* = 76) = 4.8043, = 0.028) but this significance did not hold for other personal protective equipment (e.g., owning boots, masks) or biosecurity practices (owning a footbath, having an isolation chamber for sick bords) (*p* > 0.05). Respondents were also significantly more likely (at the *p* < 0.05 level) to report using antimicrobials if their flock suffered from Newcastle disease *X*^2^ (1, *N* = 76) = 4.9108, = 0.027), Fowl Pox *X*^2^ (1, *N* = 76) = 3.9683, = 0.046), or chronic respiratory disease *X^2^* (1, *N* = 76) = 6.8672, = 0.009). Finally, respondents were more likely to report using antimicrobials if their birds displayed symptoms of snoring *X*^2^ (1, *N* = 76) = 3.9488, = 0.047).

### 3.3. Recommendations

Our analysis points to promising targets for follow-up behavioral research in order to inform future behavior change interventions to help ensure more sustainable egg production in Kiambu County—and poultry systems across LMICs—in the face of a growing threat of AMR. Given the scope of our study and in the interest of simplicity, we limit our discussion to priority behaviors around vaccination, biosecurity, and prudent use as high-priority areas for reducing AMR risks.

Below, we frame our results as priority targets for further investigation. Throughout this discussion, we draw upon our follow-up FGDs with Kiambu County poultry farmers and IDIs with animal health professionals to provide more context to our findings.

We note challenges possibly related to stakeholder knowledge (e.g. foot bath design, disinfection and cleaning methods and frequency, poultry house placement and temperature regulation, prudent use of antibiotics based on their main/true functions, lack of vaccination of indigenous species) and trust (i.e., vaccine effectiveness). We also note challenges where, despite apparently sufficient knowledge, proper vaccination, biosecurity, and antimicrobial use were not observed (e.g., disregard of vaccination schedules, insufficient frequency of cleaning drinkers or feeders, improper footbath use/replenishment). For this reason, we suggest that research on and testing of behavior change interventions be applied both in terms of training or knowledge-based actions as well as in terms of changes to decision contexts and the environment (also known as “nudges”), which target nonconscious drivers of behavior [29].

#### 3.3.1. Target #1: Improving the Salience of Information on Antibiotic Product Labels and in Agrovet Shops to Encourage Responsible AMU

It may be that some farmers and agrovets are not aware that they routinely use antimicrobials in non-prudent ways (e.g., as “egg boosters”) due to problems with the information included on the labels of antimicrobial products. As Table 1 demonstrated, while the product labels typically listed the active ingredients and the target bacteria, the same labels were missing information on the appropriate purpose of the medicine, i.e., to treat the infections for which the medicine is intended. Instead, the labels often indicated use of the medicine for nontherapeutic purposes such as prophylaxis and increased productivity. The agrovets acknowledged this and pointed out to the regulatory authority for blame. For example, one of the agrovet persons asked, “*If the medicine is wrongly labelled, how did it come to my shop? It means that it has been accepted for sale by the government*”. This presents a clear opportunity for collaboration between antimicrobial drug producers and the relevant regulatory authority to improve labeling practices, as well as an opportunity for drug dispensers to help provide corrective information on purchase.

However, research shows that product labels or instructions do not always significantly impact consumer behavior [30,31] and that the most prominently displayed marketing may influence purchasing decisions more than smaller details [32] In addition, negative warnings on drug labels have shown to be effective in grabbing attention and influencing consumer intention to purchase [33]. For these reasons, we propose examining the effect of “salience” on antimicrobial purchasing [34]. Making the dangers of antibiotics (and in particular those deemed critically important and intended to be used as a last resort) more evident through the placement of colorful or otherwise attention-grabbing symboleson product packages or in agrovet shops may help trigger behavior change. Paired with a timely reminder of the drug’s proper use or dangers from the salesperson, we suggest that this nudge intervention at the point of sale could reduce the unnecessary purchase or use of antibiotics not required by the farmer’s situation.

#### 3.3.2. Target #2: Making Vaccination Behaviors Easier to Reduce Overreliance on Antimicrobials

Our results showed the interconnectivity between a farmer’s beliefs and practices around vaccinations and antimicrobial use. As one farmer said, “*Vaccines are not preventing chickens from diseases—we doubt if they are of good quality and that is the area where we want more research done. The farmer is willing to vaccinate if we can be assured of more quality vaccines*”. If farmers lose trust in vaccinations, then they may save money that was allocated to this preventative measure and instead wait to use this money reactively on antimicrobials for treatment when diseases hit. Our qualitative work among animal health professionals also pointed to farmer practices that may be driving a poor vaccination outcome and associated doubts about vaccine effectiveness, including waiting too long during an outbreak before vaccinating. Only vaccinating upon hearing of nearby disease outbreaks points to the possibility that “hyperbolic temporal discounting” is at play—a behavioral phenomenon in which people significantly discount future gains (i.e., avoiding the loss of chickens to disease) when sacrifices in the present are necessary (i.e., paying for vaccination) [34,35].

This underscores the need for further capacity-building initiatives to improve practices such as vaccine scheduling and dosing to help promote positive health outcomes that will reinforce perceptions of vaccines as valuable infection prevention measures. It would also be helpful to investigate whether vaccine hesitancy for livestock is correlated with hesitancy towards human health vaccination initiatives as well, and whether improved management of misinformation among farmers on the effectiveness of vaccines may alter vaccine acceptance and utilization for animal and human health. Experiments around these concepts could include the provision of timely reminders via SMS or similar communication from hatcheries to farmers in addition to the standard vaccine schedules.

#### 3.3.3. Target #3: Facilitating Biosecurity Practices as a Priority for Reducing Disease Pressure Driving an Overuse of Antimicrobials

Qualitative data from producers, veterinarians, animal health assistants, and agrovets showed that most believe that poor biosecurity practices on layer farms are a primary driver of patterns of compensatory antimicrobial use. Our KAP survey results supported these views, finding limited investment in biosecurity such as footbaths, PPE, and routine use of disinfectants. Follow-up discussions with farmers suggested that these practices were not observed mainly for economic reasons—simply put, the perceived cost was greater than the perceived benefit. For example, farmers said they could not afford to keep purchasing disinfectants given the significant costs of feed and medicine. As one focus group participant stated, “*We know we should have footbaths, but we don’t have them. Most monies go to feed and medicines, so we don’t have a lot of money to concentrate on other side costs like footbaths and disinfectants—they are costly*”. This reveals a perception of biosecurity measures as optional advantages beyond the fundamental necessities of feed and drugs, rather than as necessities themselves. This finding is supported by another study in which farmers regarded the cost of AMU as the more affordable option compared to other disease management practices [36]. This suggests that discussions of risks versus benefits in the context of costs and different possible outcomes may help to shift this perspective, particularly if paired with incentives such as subsidized biosecurity supplies for farmers pledging to follow fundamental infection prevention practices as part of a local or national “antibiotic-smart” certification scheme. Such a scheme would benefit from periodic monitoring and evaluation to enable ongoing participation in the program. This approach could also open up new market opportunities and higher price points for farmer cooperatives committed to prudent antimicrobial use in response to the growing consumer demand for food produced safely and responsibly, as recently documented in a survey of East African consumers [37].

Consistent with other studies [19,36,38], we found that poor biosecurity was also driven by knowledge gaps. One farmer in our follow-up focus group told us that they (i.e., farmers in the focus group) did not think footbaths were linked to biosecurity, but instead perceived them to be part of the structural design of the chicken house. One farmer said, “*That depression at the entrance of the chicken house is not for anything, but that’s how a chicken house should be built*”. These knowledge gaps also extended to the use of PPE. Most farmers in our FGDs believed that PPE was mainly worn “*to protect me from getting dirty”* when completing their multiple daily chores, such as working in the fields and tending to the animals. As such, PPE was to protect oneself and not necessarily to prevent the spread of disease.

In addition, some farmers also shared that the design of PPE and expectations for disinfecting it made its routine use impractical. For example, some farmers reported gumboots to be particularly cumbersome (difficult to wear and remove) and very uncomfortable due to sweating of the feet. In our follow-up discussions, one of the men said, *“I wear boots in the morning and stay with them till evening. I cannot manage to keep removing and wearing them since they are difficult to remove. I have to request my wife to help take them off”.* Finally, farmers reported that they were not able to wash their PPE routinely because they were too busy with other farm activities. One woman said, “*I don’t have time to keep washing the overall and gumboots every day. I have cows that I need to search fodder for and feed, I have children to cook for, I have the chicken to attend to and go around my personal errands. I only clean them when they are very dirty*”. Therefore, discussions around PPE revealed a problematic perception of it: while protective wear may be seen as a potential advantage as a buffer against dirtying work, it is not commonly seen as a necessary or valuable precaution worth the effort to help mitigate the risk of pathogens spreading. To tackle this challenge, emotionally resonant appeals based on the serious consequences of drug-resistant infections for fellow farmers may help to shift the perceived benefit of routinely using PPE for infection prevention and help overcome the resistance associated with the perceived inconvenience. Creating social pressure through peer networks to help normalize the practice and leveraging influential community farmers to model the behavior may also help. This is also an opportunity for market-based solutions involving a diversification of PPE products better designed for comfort and long-wear for physical laborers.

Similar to the suggestion in Target #2, behavioral experiments that test nudge interventions around time preferences and temporal discounting could be implemented in addition to communication, training, and other more standard interventions. Making it more difficult to ignore footbaths via bright coloring or strategic placement could prompt more consistent use. Redesigned disinfectant measuring devices placed at strategic locations could prompt more regular and accurate replenishment. With regard to PPE, adjusting the environment in which it is stored, presented, and labeled could prompt more consistent use. Finally, messaging that associates biosecurity measures with the prevention of significant economic losses could trigger “loss aversion”, a behavioral phenomenon shown to drive actions more effectively than a focus on gains [39].

#### 3.3.4. Target #4: Facilitating Improved AMU Practices to Protect Production and Reduce Selection Pressure for AMR

Our results demonstrated that farmers displayed a variety of prudent and non-prudent AMU and AMR-related practices. A common non-prudent practice was not observing withdrawal periods. Our follow-up discussions indicated several key reasons underlying these practices. As with vaccination and biosecurity, a major reason given was the economic challenge, as has been reported in other KAP studies of African poultry systems [23,40]. One farmer, with the agreement of the entire focus group, said, *“The cost of feeding and medicines is too high and therefore if we throw away eggs, we cannot make any profits because most of us run this business with losses. After all, the medicine will be finished from the system within two days, so even if we sell eggs from birds undergoing treatment, it may not do any damage to people”*. Such practices make it more likely that antimicrobial residues enter the food supply. Interventions around triggering loss aversion, as suggested in Target #3, should be tested for their effectiveness in counteracting temporal discounting to bolster farmer understanding of the devastating effects of ignoring antimicrobial use best practices and trigger increased compliance.

Most farmers in our sample did not get prescriptions prior to obtaining antimicrobials. We found that beliefs towards prescriptions were impacted by the social norms of the role of veterinarians in animal healthcare. Our follow-up discussions indicated the veterinarian is perceived as “daktari wa ngombe”, a Kiswahili phrase literally translating to “a doctor for cows”. By viewing veterinarians as doctors of large animals, layer farmers in Kiambu may not seriously consider veterinarians (i.e., those who can give prescriptions) as options for treatment. Changing these norms will be important in promoting prudent use as our quantitative results suggest that those who seek prescriptions use less antimicrobials, possibly because the act of seeking out prescriptions may limit use (e.g., the veterinarian may conclude an antimicrobial is not needed).

### 3.4. Study Limitations

The study had several limitations. First, the Chi-squared analysis showed that many of the diseases that should drive AMU were not related to our self-report measures of use (i.e., those who reported high burdens from these diseases did not report using more antimicrobials). In part, this result suggests that multiple methods need to be developed and deployed concurrently to better characterize AMU. In this study, the use of “medicine posters”—large size posters in color with all common medicines used in poultry in the area—resulted in a diverse antimicrobial dataset. Posters were complemented by asking informants to bring used medicine containers (bottles, sachets). For future studies, passive techniques that are not as impacted by recall biases, such as collection of used antimicrobial packaging in bins, should also be used to allow for more accurate longitudinal analysis. Second, our sample size was limited to Gatundu North sub-county so we cannot generalize these statements to all Kenyan layer farmers. However, these initial assessments are needed to identify domains for further assessment and for testing methods to better understand the socioeconomic and cultural contexts of antimicrobial use and antimicrobial resistance.

## 4. Conclusions

The rising demand for poultry is generating valuable business opportunities for poultry farmers in Kenya and other LMICs. This shift includes a change in how antimicrobials are being routinely used in agricultural production. Multisectoral initiatives championing more responsible antimicrobial use and the implementation of better hygiene, sanitation, and biosecurity practices will be needed to keep antimicrobials working for as long as possible. The success of these initiatives will partially depend on effective advocacy and behavior change programs to empower key stakeholder groups to take positive action. To identify behavioral targets for these programs, a “bottom-up” understanding of AMU and AMR [19] is needed to elucidate stakeholder reasoning and context for AMU choices. Moreover, stakeholder participation in the process of understanding the main challenges and in developing solutions is more likely to incite sustainable change through a sector- and community-led approach. Here, we have shown how a mixed-methods study can identify behavioral targets by contextualizing the use of antimicrobials and how this use is framed by domains of production, cultural norms, and local veterinary infrastructures. These studies should be conducted in other production systems and different cultural/national settings to develop a comprehensive strategy to limit the emergence and spread of AMR.

## Figures and Tables

**Figure 1 antibiotics-10-00106-f001:**
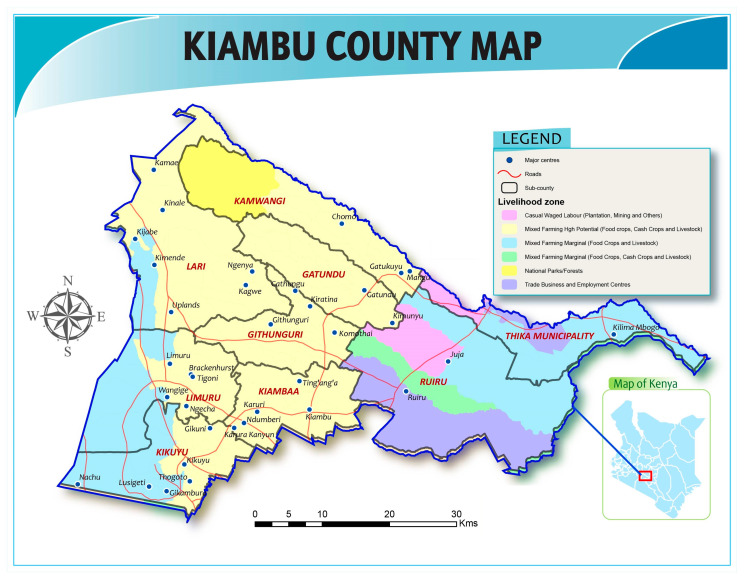
Map of study area. Compiled by FAO Kenya. Projection in Decimal Degrees with WGS 84 Datum.

**Figure 2 antibiotics-10-00106-f002:**
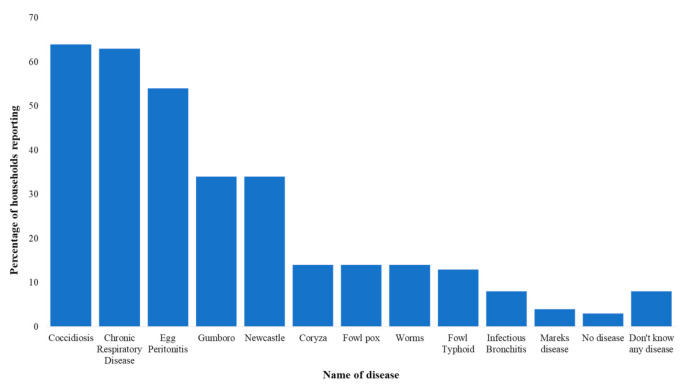
Common diseases or health conditions reported for layers in Gatundu North, Kiambu.

**Table 1 antibiotics-10-00106-t001:** Medicines reported as commonly used by layer farmers in Gatundu North identified via their chemical components, manufacturer’s description, manufacturer’s indications, and withdrawal periods. AB (1–7) refer to the antibiotics under investigation with actual trade names withheld. Only those reported by 10% or more of households are reported here. Note the manufacturer’s description of the medicine vs. the manufacturer’s indications for use, which may serve as the motivation for use. In bold are the manufacturers’ indications for use that are related to “use for prophylaxis”, while in italics are antibiotics used as “growth promoters” or to “promote productivity”.

Antibiotic Symbol	Main Chemical Components	Manufacturer’s Description on the Label	Manufacturer’s Indications for Use on the Label	Withdrawal (Eggs)	No. of HHs Reporting Use (%)
AB1	- Oxytetracycline (as HCl) - Neomycin (as sulphate) - Multivitamins	- Oxytetracycline bacteriostatic (gram-positive and gram-negative bacteria) - Neomycin bactericidal (gram-negative bacteria) - Vitamins are essential for the proper operation of several physiological functions	- *Higher peak egg production level* - *Maintenance of high production level throughout the laying period* - *Increased egg production when there is a drop-in performance caused by stress situations* - *Reduced mortality throughout the laying period* - *Increased feed conversion efficiency*	2 days	69
AB2	- Erythromycin thiocyanate - Oxytetracycline hydrochloride - Streptomycin sulphate - Colistin sulphate - Multivitamins	- Colistin bactericidal action (Gram-negative bacteria) - Oxytetracycline bacteriostatic (gram-positive and gram-negative bacteria) - Erythromycin bacteriostatic (gram-positive bacteria) - Streptomycin bactericidal (gram-negative bacteria) - Vitamins are essential for the proper operation of several physiological functions	- *Stimulates egg production, increases growth, improves feed conversion and is used as vitamin supplement during periods of diseases and stress* - Effective against gastrointestinal, respiratory and urinary infections caused by colistin, oxytetracycline, erythromycin and streptomycin sensitive micro-organisms	1 day	44
AB3	- Oxytetracycline hydrochloride	- Oxytetracycline bacteriostatic (gram-positive and gram-negative bacteria)	- Treatment of bacterial infections where the causative agents are sensitive to oxytetracycline - *In chicks: it maintains appetite during stress periods like vaccination, temperatures changes or changes in feeds*	Not shown	21
AB4	- Oxytetracycline - Neomycin - Multivitamins	- Oxytetracycline bacteriostatic (gram-positive and gram-negative bacteria) - Neomycin bactericidal (gram-negative bacteria) - Vitamins are essential for the proper operation of numerous physiological functions	- *Stimulates egg production* - *Increases growth* - *Improves feed conversion* - **Used as a vitamin supplement during periods of diseases and stress** - Treatment of gastrointestinal, respiratory and urinary infections caused by oxytetracycline and neomycin sensitive micro-organisms	1 day	19
AB5	- Tylosin tartrate - Doxycline Hcl	- Tylosin is a macrolide antibiotic and has a bacteriostatic action against gram-positive and gram-negative cocci	- Treatment of infectious coryza, CRD, *E. coli*, and cholera - **Tylo 200 wsp is very effective in the prophylaxis and therapy of Chronic Respiratory Disease (CRD) in poultry. It is also efficient in the treatment of infectious synovitis and sinusitis, as well as in several other poultry diseases.**	2 days	19
AB6	- Oxytetracycline hydrochloride - Multivitamins	- Oxysol Plus is an antibiotic/multivitamin preparation which is completely soluble in water	- Treatment and control of wide range of bacterial infections and control of secondary infections due to CRD and viral infections in chicks and all poultry - **Ideal for use in stress condition where the combination of ingredients protect the animal from debilitation caused by diseases**	1 day	14
AB7	- Oxytetracycline - Multivitamins	- Oxytetracycline bacteriostatic (gram-positive and gram-negative bacteria)	- *More eggs per kg of feed* - *Brings birds to peak production on time* - *Prevent stress and enhance production* - *Stronger eggshell through better assimilation of calcium and phosphorus even during high temperatures* - *Improves feed conversion ratio* - *Maintains laying performance* - *Promotes growth* - **Checks secondary bacterial infections during viral outbreak** - Controls infections in GIT	Not shown	10

We use italics and bold formatting to point out uses that are for growth promotion (italics) or prophylaxis (bold).

**Table 2 antibiotics-10-00106-t002:** Antimicrobial use (AMU) practices.

Practices	Never/Rarely % (N)	Sometimes % (N)	Almost Always % (N)
Antimicrobials prevent disease	26.3(20)	0 (0)	73.68 (56)
Treat all birds when one is sick	2.63 (2)	5.26 (4)	92.11 (70)
Use human medicine for birds	94.74 (72)	5.26 (4)	0 (0)
Give a larger dose than recommended	96.05 (73)	3.95 (3)	0 (0)
Give a smaller dose than recommended	96.05 (73)	3.95 (3)	0 (0)
Stop treatment early	88.16 (67)	1.32 (1)	10.53 (8)
Keep expired medicines	98.68 (75)	1.32 (1)	0 (0)

## Data Availability

Dataset and codebook is contained within the article or Appendix A.

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
