# Peer review of "Understanding Antimicrobial Use Contexts in the Poultry Sector: Challenges for Small-Scale Layer Farms in Kenya"

_antibiotics, 2021, doi:10.3390/antibiotics10020106_

Round 1

Reviewer 1 Report

The manuscript is reorganised with much better structure. The changes are highlighted in yellow, but in fact, they are mainly changes in reference numbers not in the text. Nevertheless, my comments have been considered. However, there are still several citations from farmers and vets making Results and Discussion diffuse. The personal citations are random examples of subjective opinions which not necessarily reflect the whole spectrum of results. As indicated previously, the way of discussing resembles a storytelling approach, but not a strict presentation and discussion of results in a research paper.   I suppose the citations can be tabulated in the supplementary file. In any case, the discussion must focus on the relevance of presented results.

Author Response

Dear Reviewer,

          Once again, thank you for your time and consideration of the manuscript. Please see the attached word document for our response. 

Reviewer 2 Report

How was the antimicrobial obtained by the farmers should be described and discussed.
How were the diseases diagnosed and the antimicrobial used? It is important to the disease control and AMR. 

Author Response

Dear Reviewer,

Once again, thank you for your time and consideration of the manuscript. Please see the attached word document for our response (i.e. Reviewer 2 in response table). 

Round 2

Reviewer 1 Report

Thank you for the modifications of the manuscript. My comments have been considered. 

This manuscript is a resubmission of an earlier submission. The following is a list of the peer review reports and author responses from that submission.

Round 1

Reviewer 1 Report

  1. Introduction

Line 92: for understanding (or to understand)

  1. Results

2.1

Line 107: Please take notice that min number for broilers equals 0 instead of 16 (which is actually the sd value in your Table 2). You might add “among others” or “for example” in the sentence: “Other animals kept on the farms were, among others/for example, broilers…” as some animals included in Table 2 are not mentioned in this paragraph.

2.1.1

Line 111: Focus group interviews (FGD) respondents... As “FGD” term is firstly mentioned, though it is defined in 4.2 (study design), it would be suitable to include the definition also in this line.

Line 123: “Median mortality was 50 birds…” please add the pertime interval for this mortality.

  • Vaccination Practices

Please take notice that this paragraph is written twice (lines 127-138; lines 139-149).

2.1.4 Biosecurity Practices for Infection Prevention

Line 161: Please remove “Data from FGD” after the full stop

2.1.7. AMU and AMU-Related Behaviors

From here onwards paging is wrong

Please define AMU (antimicrobial use) in Table 1 Supplemental A

Line 2: “themes/domains impacting antimicrobial use (AMU) and antimicrobial resistance (AMR)…”

  • Analyses of the Factors Associated with AMU

It would be helpful to include a Table with those variables analyzed by chi-square.

4.4. Presentation of Results and Data Analysis

Line 404: we the medicine labels: it lacks the verb

Reviewer 2 Report

The manuscript provides relevant information about the strategies regarding antimicrobial use of antibiotics in Kenya. Because of antimicrobial resistance, the use of antibiotics as performance promoters has been banned in European Union, but they are still used in several countries, including Kenya.

The results are based on interviews with producers and animal health professionals (Supplement Table 1 and 2). The interviews with “Focus group”-FGD are comprehensively described why the input from health professionals -IDI not entirely clear. The statement (L93) “qualitative interviews and quantitative knowledge attitudes and practices (KAP) surveys” is difficult to comprehend. They are further divided, in Materials, between FGDs and IDIs.

The interviews with farmers are mostly quantitative why statements from health professionals are very qualitative and general.  

Paragraph 2.1.3 is a copy of 2.1.2.

In results, there are citations of farmers and vets statements (L130, L133, L193, L209), making the results not applicable for a research article. Similarly, Discussion includes declarations making it diffuse (L264, L270, L287, L296, L305, L309, L321). A description of Results and Discussion should not be “a storytelling” approach, but a concrete outline of measured (interview) results, and then a direct discussion of their relevance.

Some AMU practices are outlined in Table 3. Although AMR reactions are described in the text, there is no outline summarising the responses, making difficult to systematically follow results.

Minor

Title: can be shortened, delete Mixed Approach.

Abbreviations shall be explained at the first occurrence, e.g. LMICs (L59), KAP (94), FGD (L111), AMU (L221). There are also several abbreviations which are only used once. These shall be omitted.

The abbreviations like FGD, IDI shall be the same, not FGDs or IDIs

L238- What is “significantly more likely”?

Table 2: “reported by 9% or more” but there is no 9%

Concluding, the presented information is valuable, but the results and discussion are not acceptable in the present form.